# Preparation and Evaluation of ^64^Cu-Radiolabled Dual-Ligand Multifunctional Gold Nanoparticles for Tumor Theragnosis

**DOI:** 10.3390/ph16010071

**Published:** 2023-01-02

**Authors:** Karim Mhanna, Wei Qian, Ziyun Zhong, Allen F. Brooks, Erika Ouchi, Jenelle Stauff, Janna Arteaga, Maria Papachristou, Ioannis E. Datseris, Bing Liu, Xia Shao, Peter J. H. Scott

**Affiliations:** 1Department of Radiology, University of Michigan, Ann Arbor, MI 48109, USA; 2IMRA America Inc., Ann Arbor, MI 48105, USA; 3College of Chemistry, Nankai University, Tianjin 300071, China; 4Department of Nuclear Medicine—PET/CT, General Hospital of Athens “Evaggelismos”, 10676 Athens, Greece

**Keywords:** PET imaging, nanoparticles, tumor imaging, copper-64, theragnostics

## Abstract

Gold nanoparticles (AuNPs) are cutting-edge platforms for combined diagnostic and therapeutic approaches due to their exquisite physicochemical and optical properties. Using the AuNPs physically produced by femtosecond pulsed laser ablation of bulk Au in deionized water, with a capping agent-free surface, the conjugation of functional ligands onto the AuNPs can be tunable between 0% and 100% coverage. By taking advantage of this property, AuNPs functionalized by two different types of active targeting ligands with predetermined ratios were fabricated. The quantitatively controllable conjugation to construct a mixed monolayer of multiple biological molecules at a certain ratio onto the surface of AuNPs was achieved and a chelator-free ^64^Cu-labeling method was developed. We report here the manufacture, radiosynthesis and bioevaluation of three different types of dual-ligand AuNPs functionalized with two distinct ligands selected from glucose, arginine–glycine–aspartate (RGD) peptide, and methotrexate (MTX) for tumor theragnosis. The preclinical evaluation demonstrated that tumor uptakes and retention of two components AuNP conjugates were higher than that of single-component AuNP conjugates. Notably, the glucose/MT- modified dual-ligand AuNP conjugates showed significant improvement in tumor uptake and retention. The novel nanoconjugates prepared in this study make it possible to integrate several modalities with a single AuNP for multimodality imaging and therapy, combining the power of chemo-, thermal- and radiation therapies together.

## 1. Introduction

Despite all the progress made in medicine, cancer is still the second leading cause of death worldwide [1]. According to an estimate provided by the World Health Organization (WHO), cancer is expected to account for approximately 10 million deaths per year by 2020 with this figure projected to rise to over 21 million by 2030, making cancer the highest mortality disease group in contrast to other diseases [2,3,4]. The cancer burden continues to grow globally, exerting tremendous physical, emotional and financial strain on individuals, families, communities and health systems [5]. Therefore, it is essential to detect this malignancy in the early stage and manage therapeutics in patients to enhance the efficiency of treatment, decrease side effects, and optimize the process for developing therapies.

Nanotechnology has been introduced into the field of biomedicine with the expectation of revolutionizing diagnostic and treatment techniques [6,7]. Nanomaterials have been used as contrast enhancement agents to promote the detection limits of diagnostic modalities [7,8]. Recently, nanotechnology has yielded considerable breakthroughs in the treatment of cancer, including chemotherapy, radiotherapy, and hyperthermia [9,10]. Taken together, multifunctional nanomaterials hold great promise for clinical theragnostic applications. Gold nanoparticles (AuNPs) are a type of widely used nanomaterial with unique chemical and physical properties. AuNPs can be readily synthesized and surface-modified with various biomolecules and anticancer drugs, making them potential theragnostic agents with high biocompatibility. In addition, AuNPs can convert optical energy into heat via nonradiative electron relaxation dynamics, which endows them with intense photothermal properties [11,12] Such localized heating effects can be directed toward the eradication of diseased tissue, providing a noninvasive alternative to surgery [13]. This photothermal therapy of AuNPs with a laser source has been established in the cancer therapy field [14]. High tumor uptake of AuNPs with great targeting ability has become the key to improving therapeutic efficiency and minimizing the side effects to normal surrounding tissue with this technique. Moreover, AuNPs have been used as contrast agents to enhance the signal of photoacoustic imaging (PAI), a noninvasive optical imaging modality that combines the merits of light and ultrasound [15,16]. Through a prior collaboration with the Xueding Wang photoacoustic laboratory, we have achieved high-resolution single cancer cell PAI imaging and dual-modality imaging with radiolabeled AuNP conjugates [17,18].

To achieve highly efficient delivery of AuNPs into cancer cells, active surface coating with functional ligands are required. Various strategies, including physical adsorption [19], thiolate chemisorption [20], and covalent coupling [21], have been developed to manipulate and modify AuNP surface chemistry. However, most methods are inadequate to achieve efficient and controllable functionalization of individual AuNPs, e.g., conjugating each AuNP with multiple types of ligands with precisely controlled amounts and distributions [22]. This is due to the fact that an excess amount of ligand is typically required to be added, with free ligand in solution helping to ensure colloidal stability of AuNPs during the conjugation, especially if AuNPs are stabilized by the charges from the adsorbed surfactants. To address this problem, we have developed a kind of colloidal AuNPs physically produced by femtosecond laser ablation of a bulk gold target in deionized water, which has a capping agent-free surface. The conjugation of functional ligands (e.g., polyethylene glycol (PEG) and peptides) to these AuNPs can be carried out with surface coverage tunable between 0% and 100%, without unreacted free ligand left in the colloidal suspension of AuNPs and maintaining the suspension stability of AuNPs is maintained [23]. In our previous work, by taking advantage of this unique property regarding surface functionalization, we coated AuNPs with quantitative amounts of both thiol-terminated polyethylene glycol (SH-PEG) molecules and cysteine-modified (Arginine(R)–Glycine(G)–Aspartic(D))_4_ (RGD) peptides. These AuNP conjugates were then selectively delivered into human prostate cancer cells. Through quantitative bioconjugation, we were able to study the cellular uptake efficiency of the functionalized AuNPs as a function of RGD peptide surface density using radioactivity analysis and photoacoustic analysis [18]. The cellular uptake efficiency of AuNP conjugates had a strong linear correlation with the surface density of RGD peptides, which supported our quantitatively controlled bioconjugation.

We have previously developed a highly efficient method to directly radiolabel AuNPs with iodine-125 [24,25,26]. The binding of ^125^I-labeled RGD-PEG-AuNP conjugates to cancer cells increased when more integrin-targeting RGD molecules were present on the AuNP surfaces [27]. The results have demonstrated our radioactivity analysis method is capable to optimize multifunctional AuNP conjugates via analyzing and evaluating their performance using various cancer cell lines. However, deiodination was observed during animal studies using I-125 labeled AuNP conjugates. To address this issue, a chelator-free Cu-64 labeling method leading to ^64^Cu-labeled AuNP conjugates with much better in vivo stability has been developed in this study. In addition to biodistribution, preclinical evaluations of ^64^Cu-labeled AuNP conjugates were also performed using PET imaging with tumor-baring mice. With the surface conjugation and radioisotope labeling technique developed, we quantitively conjugated AuNP with a combination of methotrexate (MTX) [28], a key component of cancer chemotherapy, and RGD peptide to build a multifunctional theranostic platform. Moreover, based on our experiences with [^18^F]fluorodeoxyglucose ([^18^F]FDG), a glucose analog that is the most common PET radiotracer used clinically for tumor detection, the feasibility of using a glucose chain instead of PEG was also investigated to further enhance tumor uptake of AuNP conjugates. We report here the manufacture, radiosynthesis and bioevaluation of the glucose-modified dual-ligand AuNPs conjugated with RGD and MTX for tumor theragnosis.

## 2. Results and Discussion

### 2.1. Synthesis of AuNP Conjugates

Although the initial binding mechanism of thiol with AuNPs is still under debate, thiol-modified molecules have become widely used reagents for the functionalization of AuNPs [29,30]. We have successfully conjugated AuNPs with pre-determined amounts of cysteine-modified RGD peptides and thiol-terminated PEG. The preclinical studies demonstrated the stability and biocompatibility of these AuNP conjugates, as well as their feasibility for targeted delivery [18,27]. Thus, a thiol-terminated PEG chain was chosen as the linker to connect the bioactive ligands onto the surface of the AuNPs.

To enhance the therapeutic function of the AuNPs, we next conjugated chemotherapy agent MTX onto AuNPs using the PEG molecules as stabilizing ligands. MTX is a structural analogue of folic acid and is one of the most widely used antimetabolites in cancer chemotherapy [31]. It is also a part of the established treatment of autoimmune inflammation [28]. Moreover, ^99m^Tc-MTX has been used as an imaging probe for the detection of tumors and sites of inflammation, and MTX derivatives have also recently been introduced into multifunctional nanoparticles for theragnosis [32,33]. To achieve the conjugation of AuNPs with MTX, it was attached to the terminal amine of thiol-PEG-amine molecules via an amide linkage to form MTX-PEG conjugates as shown in Figure 1 [34,35]. Since the conjugation reaction gave 85% of γ-derivative under anhydrous conditions, protection of α-the carboxylic group was not necessary. The total unoptimized yield was 21%.

[^18^F]FDG is a glucose analog and has been the most popular tumor detection PET radiotracer for decades, given the increased uptake and metabolism of glucose in tumors. Carbohydrate-coated or sugar-based nanoparticles have been used to increase the local concentration of imaging probes and enhance the detection signal of tumors [36,37,38]. To prepare glucose-modified AuNPs, thiolated PEG coupled with gluconic acid d-lactone as shown in Figure 2. The total unoptimized yield was 6%.

The raw AuNPs were fabricated using our patented technology [39], femtosecond pulsed laser ablation (PLA) of a bulk gold target submerged in flowing deionized water. The PLA method was chosen for this work because compared to others, typically chemical methods, it is a complete green synthesis of AuNPs in deionized (DI) water. This method uses tightly focused micro-joule (μJ) femtosecond laser pulses to produce NPs. The average size and size distribution of generated NPs can be precisely controlled by optimizing laser parameters, such as wavelength, pulse energy, duration, and repetition rate, etc. [40]. Colloidal AuNPs with an average diameter of 20 nm and zeta potential of −43 mV were produced and used in the experiments. Representative transmission electron microscopy (TEM) image, size distribution, and ultraviolet–visible (UV–vis) absorption spectrum of the generated AuNPs are presented in Figure 1. The generated nanoparticles have a narrow size distribution and an absorption peak at 520 nm due to the localized surface plasmon resonance (LSPR) [41]. The spectral feature below 450 nm reflects gold intraband transitions since the nanoparticles were generated in deionized water. In addition, AuNPs produced using the PLA method are naturally negatively charged without requiring any stabilizing/capping ligands for maintaining their colloidal stability [42]. Therefore, they have a capping agent-free surface. This unique feature allows versatile and controllable surface manipulation and modification, which serves as the foundation for quantitative bioconjugation of the PLA-generated colloidal AuNPs.

Following the methodology developed in our previous work [18,23], the raw colloidal AuNPs were quantitatively functionalized with two different types of ligands, i.e., PEG molecules and RGD peptides for effective and selective targeting of cancer cells. The functionalization was performed in a sequential manner by first mixing the colloidal AuNPs with the 1st ligand and then the 2nd ligand, both of which are terminated with thiol groups (SH). The quantification was achieved by precisely controlling the molar ratio of the ligands to the colloidal AuNPs, i.e., PEG/AuNPs and RGD/AuNPs. PEG chains grafted onto the AuNPs can improve their stability and biocompatibility, and simultaneously minimize nonspecific interactions with biological tissues. The binding of thiol-terminated molecules to the AuNPs is possible due to the strong anchoring of the thiol-gold bonds, which covalently attach the ligands to the surfaces of the nanoparticles. The conjugation of two types of ligands onto AuNPs in a sequential manner could be experimentally monitored via dynamic light scattering (DLS) measurements since the hydrodynamic diameters of AuNPs would increase as more and more ligands bind onto the surface of AuNPs. In this study, we have fabricated a series of AuNP conjugates shown in Table 1. Based on our previous results [27], the highest loading of 1800 molecules on each 20 nm AuNP was used with 900 molecules of each ligand. The increase in the hydrodynamic diameters during the fabrication of these AuNP conjugates in a sequential manner was observed and reported in Table 2, which confirmed the binding of the ligands as well as quantitatively controlled bioconjugation. In addition, the AuNPs conjugated with methoxy-PEG-SH were used as control.

### 2.2. Cellular Uptake of AuNP Conjugates Labeled with Iodine-125

To screen the fabricated AuNP conjugates, they were first labeled with ^125^I (since ^64^Cu is expensive and not available locally, we reserved this for in vivo work as discussed below) and evaluated in vitro using HeLa cells, because this cell line was found to be remarkably durable and prolific, has high expression of folate cells and is the most commonly used human cancer cell line in scientific research [43]. The same procedures were applied as previously reported [27]. Briefly, the AuNP conjugates were labeled with ^125^I and then incubated with a monolayer (~75% confluence) of HeLa cells at 37 °C for 12–15 h. The cells were separated from media and rinsed with cold PBS. The percentage of the radioactivity retained in the cells due to internalized AuNP conjugates through endocytosis was calculated as the cellular uptake.

In order to increase the colloidal stability and biocompatibility of AuNPs in biological environments, PEG molecules have been widely used to graft onto the nanoparticle surface. The PEG layer can shield the AuNPs from fouling by serum proteins and reduce their rate of clearance by the reticuloendothelial system (RES) of the body [42], but PEG molecules, especially for those with longer chains, may block the binding sites of the targeting ligands, such as RGD and MTX. Thus, a PEG chain needs to be long enough to stabilize nanoparticles in vivo but not too long to block the targeting moieties. Based on our previously reported cellular data [27], the PEG-RGD-AuNP conjugates (**1**) with PEG 2000 (molecular weight = 2000) and 5000 (molecular weight = 5000) were tested for comparing their cellular internalization performance. The cellular uptake of conjugates was 14.7 ± 0.5% and 8.7 ± 0.4% for PEG 2000 and 5000, respectively. The longer PEG chain shielded the RGD from binding to the receptors on the plasma membrane of cancer cells and reduced its integrin-targeting ability, resulting in less cellular recognition and uptake. PEG 2000 were then used for all further experiments, including the PEG attached to the MTX and glucose.

The cellular uptake of novel MTX-conjugated AuNPs (AuNP conjugate **2**, PEG-MTX-AuNP) was 6.4 ± 0.9%, while the negative control sample of AuNPs containing only mPEG on the surface (PEG-AuNP) showed very little cellular uptake (0.10 ± 0.02%). This confirmed the targeting capability of MTX via the folate transport system [44]. Nanoparticles surface conjugated with multiple targeting ligands have been proven to promote intracellular delivery and therapeutic efficacy [33]. Therefore, we prepared AuNPs conjugated with the combination of RGD and MTX (MTX-RGD-AuNP, conjugate **4**). However, the cellular uptake of this conjugate (**4**) was 12.1 ± 1.1%, higher than that of MTX nanoconjugates (**2**) as expected, but lower than that of RGD nanoconjugates (**1**). The cytotoxic effect of MTX is under investigation and will be reported in course.

### 2.3. Chelator-Free Cu-64 Labeling

In our previous work, we have developed a highly efficient method to directly radiolabel AuNP conjugates with I-125 radioisotope and successfully evaluated their cellular uptake. However, deiodination was observed during animal studies using I-125-labeled AuNP conjugates. The desirable nuclear properties and straightforward radiolabeling chemistry make Cu-64 (t_1/2_ = 12.7 h, β^+^ 653 keV [17.5%], β^−^ 597 keV [38.5%]) the most widely used positron emitter for nanoparticle-based molecular imaging. In addition, its therapeutic effects due to its β^-^ radiation of cancer cells have attracted attention recently [45]. Therefore, we used ^64^Cu-labeled AuNP conjugates for in vivo studies.

A method of reducing ^64^CuCl_2_ by hydrazine (N_2_H_4_) in the presence of poly(acrylic acid) for Cu-64 radiolabeling has been reported. By using this method, the trace amount of Cu-64 could be integrated and epitaxially grown onto the surface of AuNPs. The procedures were simple, and the labeling yields were high. However, extra efforts in purification and quality control are required for animal studies due to the high toxicity of hydrazine. Thus, we modified this method by using ascorbate as a reducing agent instead of hydrazine. Briefly, sodium ascorbate was dissolved in borate buffer (pH 8.0) and added into AuNPs in water, followed by ^64^CuCl_2_ aqueous solution. The mixture was allowed to react at room temperature for 1 h before washed with water three times by centrifugation. The radiochemical yields were 54–83%, comparable with the previously reported method. The final products were stable with no visible aggregation in sterile water (pH 7.4) and PBS (pH 5.5) at room temperature after 2 and 5 days, respectively because their color was the same as that of the original colloidal AuNPs. This improved method produced high yield, reliable and stable ^64^Cu-labeled AuNP conjugates, which are safer for biomedical applications via replacing hydrazine with biologically compatible sodium ascorbate. This modified ^64^Cu-radiolabeling method will also benefit clinical translation in the future.

### 2.4. Small-Animal PET Imaging and Biodistribution

The tumor-bearing nude mice were used in PET imaging studies. About 150 µCi of ^64^Cu-AuNPs was injected into each animal (i.v., tail vein). The mice were imaged dynamically (10 × 0.5 min, 5 × 5 min, 3 × 10 min) for 60 min at 22- (approx. 1 day) and 45 h (approx. 2 days) post-injection (Figure 2 and Figure 3, respectively). Regions of interest (ROIs) were drawn over the tumor and muscle and used to calculate tumor-to-muscle ratios (Table 3) and SUV max (Table 4). As shown in Figure 2 and Figure 3, the tumor uptake of dual ligand AUNPs **4**, **5**, and **6** was higher than that of the single-ligand AUNPs **1**, **2** and **3**. This is in marked contrast to PET imaging of free ^64^CuCl_2_, which shows rapid renal clearance (data not shown), and imaging of non-conjugated AuNPs which showed most activity in the liver within 5 min [24]. Furthermore, the glucose-modified AuNP conjugates **5** and **6** showed significant improvement of tumor uptake and retention, especially at the late time point (45 h) after background signal clearance from other organs. The replacement of mPEG with the glucose-modified PEG maintained the stability and biocompatibility of AuNPs, and also increased targeted delivery of the NPs to the tumor. To further confirm the accumulation of AuNPs in the tumor, biodistribution data (Figure 4) were obtained by sacrificing the mice after PET imaging with the most promising AuNPs (**4**–**6**). ^64^Cu-RGD-MTX-AuNP conjugate **4**, ^64^Cu-RGD-Glucose-AuNP conjugate **5** and ^64^Cu-Glucose-MTX-AuNP conjugate **6** each showed tumor uptake lower than that of liver but higher than other organs. These nanoconjugates that combined the PET imaging probe with a chemotherapy drug have the potential for cancer theragnosis. In addition, the accumulation of AuNPs could enhance the contrast of optical imaging [16] and enable photothermal therapy by exposing tumor site identified by PET imaging to laser irradiation. Moreover, the β^−^ radiation of bound ^64^Cu has been demonstrated recently [46], offering the potential for radiation therapy applications as well. It is also feasible to attach other therapeutic isotopes onto the AuNPs using our controllable bioconjugation technique. Thus, this unique nano-platform offers a versatile tool for multimodality imaging and therapy methods: chemo-, thermal- and radiation therapies, although we recognize the latter application will only be possible if hepatic uptake can be substantially reduced so as to avoid liver damage from high doses of therapeutic radionuclides. The testing of the therapeutic efficacy of various dual-ligand AuNPs conjugates is underway, including further investigation of the effects of modifying our AuNP conjugates, and evaluation of their in vivo toxicity.

## 3. Materials and Methods

### 3.1. Chemicals and Materials

All chemicals were used as received without purification. Thiol-terminated PEGs were purchased from Creative PEGWorks. Cysteine-modified (RGD)_4_ was custom-synthesized by RS synthesis LLC. D-(+)-Gluconic acid d-lactone was purchased from Sigma-Aldrich, St. Louis, MO, USA. (2*S*)-2-[[4-[(2,4-diaminopteridin-6-yl)methyl-methylamino]-benzoyl]amino]pentanedioic acid (MTX) was purchased from Supelco, Bellefonte, PA, USA. [^125^I]Sodium iodide aqueous solution was purchased from PerkinElmer Inc., Waltham, MA, USA. [^64^Cu]copper chloride aqueous solution was purchased from Washington University in St. Louis, MO, USA.

### 3.2. Synthesis of MTX-PEG-SH

0.1 g (0.22 mmol) of (2*S*)-2-[[4-[(2,4-diaminopteridin-6-yl)methyl-methylamino]-benzoyl]amino]pentanedioic acid (MTX) and 0.5 mL of triethylamine were added to 4 mL of DMSO and stirred until completely dissolved. To this solution, 0.025 g (0.22 mmol) of N-hydroxy succinimide (NHS) and 0.05 g of dicyclohexylcarbodiimide (DCC) were added and the reaction mixture was stirred under nitrogen atmosphere overnight at room temperature. The formed precipitate was removed by filtration and the filtrate was poured into 25 mL of cold ethyl acetate. The subsequent precipitate was collected by filtration and washed with cold methanol. The yellow powder was dissolved into 10 mL of DMSO and 0.2 mL of triethylamine. To the solution was added 0.4 g (0.2 mmol) of thiol-PEG-amine (molecular weight = 2000). The reaction mixture was stirred at room temperature for 48 h and then poured into cold ethyl acetate. The precipitate was collected by filtration and re-dissolved into water. The aqueous solution was filtered and lyophilized to dryness. 0.105 g of light-yellow powder was obtained. Calculated MS: 2452. MALDI MS: 2403—2887.

### 3.3. Synthesis of Glucose-PEG-SH

0.1 g (0.56 mmol) of D-(+)-Gluconic acid d-lactone was dissolved in 10 mL of methanol and 0.2 mL of triethylamine. The solution was added to 1 g (0.5 mmol) of thiol-PEG-amine (molecule weight = 2000). The mixture was stirred at room temperature for 18 h and evaporated in vacuo to half the volume. Acetone was then added to give a precipitate. The precipitate was collected by filtration and re-dissolved into water. The aqueous solution was filtered and lyophilized. 0.06 g of light-yellow powder was obtained. Calculated MS: 2176. MALDI MS: 2005–2312.

### 3.4. Physical Production of Colloidal AuNPs

The ytterbium-doped femtosecond fiber laser (FCPA μJewel D-1000, IMRA America, Ann Arbor, MI, USA) operating at 1.045 μm delivered pulsed laser at a repetition rate of 100 kHz with 10 μJ pulse energy and 700 fs pulse duration. The emitted laser beam was first focused by an objective lens and then reflected by a scanning mirror to the surface of the bulk gold target (16 mm long, 8 mm wide, 0.5 mm thick, and 99.99% purity, Alfa Aesar, Ward Hill, MA, USA), which was submerged in flowing deionized water (18 MΩcm). The size of the laser spot on the gold target was estimated to be 50 μm and its position was precisely controlled by the scanning mirror. During the ablation process, a translation stage was employed to produce relative movements between the laser beam and the gold sample. The generated nanocolloids were stably suspended in water and did not require any dispersants, surfactants or stabilizers to maintain their stability. The produced colloidal AuNPs were characterized by an array of analytic instruments and techniques, including TEM, DLS, and UV–vis) spectroscopy. Images of the colloidal AuNPs were recorded using a TEM (JEOL 2010F, Tokyo, Japan) at an accelerating voltage of 100 kV. Nanoparticle hydrodynamic diameter size and zeta potential were characterized via DLS analyses using a Nano-ZS90 Zetasizer (Malvern Instrument, Westborough, MA, USA). UV–vis absorption spectra were recorded by a spectrophotometer (UV-3600, Shimadzu Corp., Kyoto, Japan).

### 3.5. Quantitatively Controlled Conjugation of AuMPs

A sequential conjugation procedure was followed to quantitatively graft functional ligands onto the surface of the colloidal AuNPs. The functionalization process comprises two steps.

Step 1. PEGylated the colloidal AuNPs (average diameter 20 nm, optical density or OD 1) by mixing AuNP with the PEG-SH solution or MTX-PEG-SH or Glu-PEG-SH. The molar ratio of the PEG solution to the colloidal AuNP was tuned to 900 (i.e., PEG/AuNPs = 900). The mixture was allowed to stand for 2 h at room temperature to enable sufficient conjugation of PEG with the AuNPs via thiol-gold bonding. The resultant colloidal solution of AuNPs was centrifuged (5000 g, 10 min) twice with supernatants discarded. The final PEGylated AuNPs were collected and resuspended in deionized water to an OD of 20.

Step 2. Conjugated the PEGylated AuNPs (OD 20) with the 2nd ligand by mixing it with RGD peptides or MTX-PEG-SH. The number of 2nd ligand conjugated on the surface of the AuNPs was quantitatively determined by controlling the molar ratio of the ligand to the AuNPs (e.g., RGD/AuNPs = 900). The resultant solutions were allowed to stand for 24 h at room temperature. The final solutions were again centrifuged (5000 g, 10 min) twice with supernatants removed. The resultant AuNPs were collected and resuspended in water.

The quantitatively controlled conjugation of the AuNPs was confirmed by measuring their hydrodynamic diameters and zeta potentials during the conjugation by using DLS analyses.

### 3.6. Preparation of ^64^Cu-Labeled AuNP Conjugates

1 mg of sodium ascorbate was dissolved in 150 uL of 0.1 M borate buffer (pH 8.0) and then mixed with 400 uL of AuNPs (20 nm, OD 10). To this solution, 400 uCi of ^64^CuCl_2_ aqueous solution (>6405 mCi/umoL) was added. The mixture was allowed to react at room temperature for 1 h. The reaction solution was centrifuged at 3000 g, for 15 min with supernatants discarded. The AuNPs were resuspended in 500 uL DI water and washed with DI water twice by centrifugation. The radiochemical yields ranged from 54–83% (n = 8). Radiochemical purity >97%.

### 3.7. Cellular Uptake of AuNPs

The HeLa 229 cell line was obtained from the American Type Culture Collection (ATCC) and maintained in Dulbecco’s Modified Eagle Medium (DMEM) supplemented with fetal bovine serum (10%). The cells were cultured at 37 °C in a humidified atmosphere containing 5% CO_2_. About 30 h prior to the experiment, the cells were transferred into 6-well plates. At desired confluences, approximately 1 million cells per well, I-125 labeled AuNPs were added into the cells, 3–5 uCi per well. The cells were then incubated for 12–15 h at 37 °C. The media containing non-binding nanoparticles were removed and the cells were rinsed with 1 mL cold PBS buffer three times to wash off excess nanoparticles. The washed cells were then treated with 1 mL 0.05% trypsin-ethylenediaminetetraacetic acid, (Thermo Fisher Scientific, Waltham, MA, USA, 25200056) solution for 10 min to ensure detachment from the bottoms of the wells for subsequent radioactivity analysis. The level of radioactivity of the cells and of each removed solution was then measured using an Auto-Gamma counter 5000 (Packard, Downers Grove, IL, USA). The percentage of radioactivity in cells over the total radioactivity for each well was calculated.

### 3.8. Small-Animal PET Imaging and Biodistributions

All animal experiments were conducted under the supervision of the University of Michigan Institutional Animal Care and Use Committee (IACUC) and were conducted according to approved protocols in accordance with all applicable federal, state, local and institutional laws or guidelines governing animal research.

The HeLa 229 cell line was obtained from the American Type Culture Collection (ATCC) and maintained in Dulbecco’s Modified Eagle Medium (DMEM) supplemented with fetal bovine serum (10%). The cells were cultured at 37 °C in a humidified atmosphere containing 5% CO_2_. At about 50% confluence, approximately 5 million cells were injected into 6- to 8-week-old nude mice (Charles River, Wilmington, MA, USA). Mice were housed under pathogen-free conditions in micro-isolator cages, with rodent chow and water available ad libitum. Animals were examined daily, and body weight and tumor sizes were determined. Tumors were allowed to grow to a diameter of 4–6 mm (short axis) before imaging. Nude mice bearing tumor received 80 uL of ^64^Cu-AuNPs (120–160) μCi intravenously and microPET imaging was performed at 22 and 45 h post-injection. Mice were anesthetized with isoflurane and maintained on 1% isoflurane throughout the imaging period. The animals were placed on their dorsal side in the scanner, and body temperature was maintained using a heating pad. Following a measured transmission scan, and a 60 min dynamic scan was performed with Concorde Microsystems P4 PET scanners.

After the last scan, the animals were sacrificed at 46 h post-injection. Tissue samples were removed and weighed, and the radioactivity was measured in a gamma counter. The uptake of AuNPs in organs was calculated as a percentage of the injected dose per gram of tissue.

## 4. Conclusions

We have designed and fabricated gold nanoparticles precisely conjugated with different types of functional ligands having pre-determined amounts. Both in vitro and in vivo studies demonstrated that tumor uptakes of dual-ligand AuNPs were higher than that of single-ligand AuNPs. The glucose-modified AuNP conjugates showed significant improvement in both tumor uptake and retention. While further work is required to establish safety and efficacy, the novel nanoconjugates described in this study offer future opportunities to combine multimodality imaging and therapies together into one nano construct with potential applications in cancer theragnosis. Further optimization of the nanoconjugates with libraries of bioactive molecules and evaluation of therapeutic potential are both underway.

## Data Availability

Data are contained within the article.

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
