# Peer review of "Preparation and Evaluation of 64Cu-Radiolabled Dual-Ligand Multifunctional Gold Nanoparticles for Tumor Theragnosis"

_pharmaceuticals, 2023, doi:10.3390/ph16010071_

Round 1

Reviewer 1 Report

This manuscripts reports the 64Cu labeled AuNPs modified with different combination of PEG,RGD,glucose-PEG-SH, or MTX-SH RGD for tumor theraonistics. The study includes nanoparticles synthesis, characterization, radiolabeling, in vivo imaging and evaluation. Overall, the study is interesting and novel. It can be accepted for publication after addressing the following concerns.

1)      MTX-PEG-SH and Glucose-PEG-SH were prepared for modification of AuNPs. Only measured the MW was provided. It would be helpful to give more data to support the successful synthesized the compounds such as expected MW, purity, ect.

2)      Any data on the Zeta potential of the Au- nanoparticles prepared?

3)      It is not clear the purity of 64Cu-AuNPs injected and stability of 64Cu labeled AuNPs. Free 64Cu as impurity or released from nanoparticles may accumulate in tumor and compromise the interpretation of the imaging results.

4)      No specificity to verify the targeting specificity. This should be discussed at least.

5)      In table 2, provide the SD values for the T/M ratio and statistical analysis of the data.

6)      Is there any data on the quantitative analysis of PET images to show the tumor and normal organ uptakes?

7)      The study does not show any therapeutic data. The conclusions on therapeutic potential of the nanoparticles should be carefully revised.

Author Response

This manuscripts reports the 64Cu labeled AuNPs modified with different combination of PEG,RGD,glucose-PEG-SH, or MTX-SH RGD for tumor theraonistics. The study includes nanoparticles synthesis, characterization, radiolabeling, in vivo imaging and evaluation. Overall, the study is interesting and novel. It can be accepted for publication after addressing the following concerns. 

  • MTX-PEG-SH and Glucose-PEG-SH were prepared for modification of AuNPs. Only measured the MW was provided. It would be helpful to give more data to support the successful synthesized the compounds such as expected MW, purity, ect. 

We greatly appreciated the reviewer’s comment. As the reviewer recommended, in the experimental section of the revised manuscript, we have added the expected MW for both MTX-PEG-SH (line 342, page 9) and glucose-PEG-SH (link 349, page 9). Since the raw thiol-PEG-amine purchased has a wide range of molecular weight, the measured MW was reported as a range not an exact MW.

  • Any data on the Zeta potential of the Au-nanoparticles prepared?

We thank the reviewer for this valuable question. We have added the zeta potential of the Au nanoparticles prepared, which is around -43 mV, in the revised manuscript (line 149, page 4).

3)      It is not clear the purity of 64Cu-AuNPs injected and stability of 64Cu labeled AuNPs. Free 64Cu as impurity or released from nanoparticles may accumulate in tumor and compromise the interpretation of the imaging results. 

      We greatly appreciate the reviewer for pointing this out. In our fabrication of AuNPs labeled with 64Cu. The 64Cu-AuNPs were washed with deionized water twice to remove free 64Cu. We have tested the efficiency of the washing process and found that the activity recovered from 1st and 2nd wash were 88±0.4% and 97±0.02%, means greater than 97% of activity were attached on nanoparticles. We have added in the revised manuscript the radiochemical purity of 64Cu-AuNPs (line 398, page 10).

      In addition, PET imaging of free 64CuCl2 showed rapid renal clearance (not reported data). Gamma imaging of no-conjugated AuNPs showed the most activity locked in live within 5 minutes (ref 25, our previous report).  This is also reflected in the manuscript.

4)      No specificity to verify the targeting specificity. This should be discussed at least. 

      We greatly appreciate the reviewer for this comment. The ligands we selected to make gold nanoparticle conjugates are all well-known for their specificity in targeting tumor cells. Although we did not perform blocking studies, our negative control tests using AuNPs conjugated with mPEG only showed the cellular uptake as low as 0.10±0.02% (line 215, page 6). In addition, we have provided in the revised manuscript serval references for RGD (19, 28), MTX (32-34), and glucose (37), demonstrating their high specificity in targeting tumor cells.

5)      In table 2, provide the SD values for the T/M ratio and statistical analysis of the data. 

      We thank the reviewer for this comment. We have added the SD values in the table 3 (page 7).

6)      Is there any data on the quantitative analysis of PET images to show the tumor and normal organ uptakes?

      We greatly appreciate the reviewer for this valuable question. We have added the SUV max of tumor in the table 4 (page 8). Biodistribution were performed to compare the uptakes of AuNP conjugates between tumor and normal organs (figure 4, page 8)

7)      The study does not show any therapeutic data. The conclusions on therapeutic potential of the nanoparticles should be carefully revised

      We thank the reviewer for this comment. The conclusions have been revised.

Reviewer 2 Report

Review of manuscript pharmaceuticals-2000711

The manuscript describes the preparation and evaluation of 64Cu chelator free-radiolabelled Gold nanoparticles coated with either Glucose or PEG  containing  RGD and/or Methotrexate for tumor theragnostic. The concept is very interesting, and if proven successful may aid substantially the efficacy in the treatment of cancer. The manuscript is generally well-written and easy to follow. However, there are several issues that in my opinion should be addressed.

·         Page 4 line174: The authors state “The binding, as well as quantitatively controlled bioconjugation, were experimentally confirmed by dynamic light scattering (DLS) measurements” more than once in the manuscript but these measurements are missing from the manuscript. Please add.

·         Page 5 line 182: “The fabricated AuNP conjugates were first evaluated in vitro using HeLa cells, the most commonly used human cancer cell line”. The most commonly used human cancer cell line in what? Why?. Please explain.

·         The cellular uptake was performed only in the PEG-AUNPs and not on the Glu-AuNPs labeled with a halogen (125I) and not the 64Cu metal which can produce different properties including different charges. The manuscript is about 64Cu-AuNPS therefore,  the cellular uptake needs to be of the molecules that were also used for the in vivo PET/Bio-distribution study so a direct correlation can be achieved.

·         The authors also chose 2 different sizes of PEG (2k and 5k). Please specify based on what were these 2 sizes chosen.

·         Page 6 line229: “The final products were stable with no visible aggregation in sterile water (pH 7.4) and PBS (pH 5.5) at room temperature after 2 and 5 days, respectively.” How was this stability measured? And just because there aren’t visible aggregates that don’t mean that they are not there. Why a TLC/ITLC was not performed to detect aggregates?

·         Page 6 line241: The authors state:” As shown in Figure 2, the tumor uptake of dual-ligand AuNPs 3 was higher than that of single-ligand AuNPs 1 and 2.”However, according to the cations in Figure 2 and Table 2, AuNPs 1 were not tested which is very strange taking into account that in the in vitro study it was the one with the most uptake in cells. Please check.

·         Page 6 line245:” The replacement of mPEG with the glucose-modified PEG maintained the stability and biocompatibility of AuNPs…”How was this confirmed??

·         Page 6: The authors mention that the 64Cu-AuNPs could also be used in radiation therapy as well. With so high liver uptake is this feasible?

·         Page6: Biodistribution of 64Cu-AuNPs 1 and 2 are missing.

·         Page 9: Each ligand was added in a ratio of 900 to AuNPs. How it is guaranteed that both ligands are bound equally to the NPs?

·         Page 9 section 3.6: The QC of the product is missing as well as its radiochemical purity.

·         Page 10 line 380: The PET scan times of 22 and 45 hours were based on what?

·         Page 10 line 392: “The modified radiolabelling method has proven to be highly reliable and presents benefits for potential clinical translation in multimodality imaging or theragnosis of cancer.” Please specify which labeling method do you refer to.

·         Page 10 line 393:”Both in vitro and in vivo studies demonstrated that tumor uptakes of dual-ligand AuNPs was higher than that of single-ligand AuNPs”. I do not agree with the statement since no statistical analysis was shown. Please reformulate your conclusion more accurately.

·         In my opinion one crucial compound, 64Cu-AuNPS, is missing in the in vivo study since the HeLa 229 xenografts are known to be highly vascularized there is no way to prove that this increased uptake is not prefusion mediated.

Author Response

The manuscript describes the preparation and evaluation of 64Cu chelator free-radiolabelled Gold nanoparticles coated with either Glucose or PEG  containing  RGD and/or Methotrexate for tumor theragnostic. The concept is very interesting, and if proven successful may aid substantially the efficacy in the treatment of cancer. The manuscript is generally well-written and easy to follow. However, there are several issues that in my opinion should be addressed.

  • Page 4 line174: The authors state “The binding, as well as quantitatively controlled bioconjugation, were experimentally confirmed by dynamic light scattering (DLS) measurements” more than once in the manuscript but these measurements are missing from the manuscript. Please add.

We greatly appreciated the reviewer for this comment. As the reviewer recommended, in the revised manuscript, we have added the results of increase in hydrodynamic diameter determined by DLS measurements (Table 2, page 5). Also, we have added a discussion on confirming binding of ligands as well as quantitatively controlled bioconjugation via characterizing size increase by DLS measurements.

  • Page 5 line 182: “The fabricated AuNP conjugates were first evaluated in vitro using HeLa cells, the most commonly used human cancer cell line”. The most commonly used human cancer cell line in what? Why?. Please explain.

We thank the reviewer for this question. We have explained in the revised manuscript that the HeLa cells are the most commonly used cancer cell line in scientific research because they are remarkably durable and prolific (lines 192-194, page 5). We have also provided a reference (44) for this explanation.

  • The cellular uptake was performed only in the PEG-AUNPs and not on the Glu-AuNPs labeled with a halogen (125I) and not the 64Cu metal which can produce different properties including different charges. The manuscript is about 64Cu-AuNPS therefore,  the cellular uptake needs to be of the molecules that were also used for the in vivo PET/Bio-distribution study so a direct correlation can be achieved.

We greatly appreciate reviewer’s comment. We performed cellular experiments using I-125 because its longer half-life and easier accessing. The 64Cu is expensive and not available locally, so it’s only used for animal studies. This is now reflected in the manuscript.

  • The authors also chose 2 different sizes of PEG (2k and 5k). Please specify based on what were these 2 sizes chosen.

We thank reviewer for this comment. We have added in the revised manuscript an explanation that a PEG chain needs to be long enough to stabilize nanoparticles in vivo but not too long to block the targeting moieties (lines 203-205, page 5). In addition, based on our previously reported data (ref 25), both 2k and 5k PEGylated nanoparticles were used in initial cell experiments. The results showed that the longer 5k PEG chain shielded the RGD from bonding to the receptors on the cancer cells and reduced integrin-targeting ability of AuNPs, resulting in less cellular uptake (lines 205-206, page 5). Thus, 2k PEG were used for all further experiments (line 212-213). 

  • Page 6 line229: “The final products were stable with no visible aggregation in sterile water (pH 7.4) and PBS (pH 5.5) at room temperature after 2 and 5 days, respectively.” How was this stability measured? And just because there aren’t visible aggregates that don’t mean that they are not there. Why a TLC/ITLC was not performed to detect aggregates?

We thank the reviewer for this valuable question. We have updated the manuscript as “The final products were stable with no visible aggregation in sterile water (pH 7.4) and PBS (pH 5.5) at room temperature after 2 and 5 days, respectively because their color was the same as that of the original colloidal AuNPs” (lines 243-245, page 6). Colloidal stability of Au NPs could be visually determined because of their unique optical property stemming from the localized surface plasmon resonance (LSPR). For example, for Au NPs with diameter of 20 nm, similar as those used in our studies, their LSPR is around 520 nm, which make colloidal solution of dispersed Au NPs appear to be pink in color. If aggregation of Au NPs occurs in solution, the LSPR will be red-shifted which causes a color change from pink to purple and to blue eventually. Since the red-shift of LSPR is very sensitive to the aggregation of Au NPs, a color change of the colloidal Au NPs could be observed, even there are only tiny amount of Au NP aggregates. Thus, the color of colloidal Au NPs provides an extremely sensitive way for visually determining whether or not there is aggregation of Au NPs.

We greatly appreciate the reviewer for suggesting of using TLC/ITLC to detect aggregates. For carrying out this measurement, we have to take final sample of Au NP conjugates outside of lab. However, we could not do it due to radioactivity of the final Au NP conjugates. We will consider radio-TLC for future studies.

  • Page 6 line241: The authors state:” As shown in Figure 2, the tumor uptake of dual-ligand AuNPs 3 was higher than that of single-ligand AuNPs 1 and 2.”However, according to the captions in Figure 2 and Table 2, AuNPs 1 were not tested which is very strange taking into account that in the in vitro study it was the one with the most uptake in cells. Please check. 

We greatly appreciate the reviewer for this comment. In the revised manuscript, we have added PET images for both  AuNP conjugates 1 (RGD only) and AuNP conjugates 3 (Glucose only) (figure 2, page 7). In addition, we have added the SUV max of tumor (table 4, page 8). We did not record PET images at 45 hrs post-injection for both AuNP conjugates 1 (RGD only) and AuNP conjugates 3 (Glucose only) due to the low tumor uptake shown at 22 hrs post-injection. That is why we did include those images initially but we have added them to the revised manuscript.

  • Page 6 line245:” The replacement of mPEG with the glucose-modified PEG maintained the stability and biocompatibility of AuNPs…”How was this confirmed??

We thank the reviewer for this valuable question. Our statement that the replacement of mPEG with the glucose-modified PEG maintained the stability and biocompatibility of Au NPs was confirmed by not observing visible aggregation during manufacture and radiolabeling of glucose-modified AuNPs. This is now reflected in the manuscript.

  • Page 6: The authors mention that the 64Cu-AuNPs could also be used in radiation therapy as well. With so high liver uptake is this feasible?

We thank the reviewer for pointing this out. Gold has been used as medicine for hundreds of years. Its metabolic mechanism is complicated and has not been fully understood yet. However, it has been known that coating AuNPs with PEG definitely reduces their toxicities and increases biocompatibility. Also, further modification of our Au NP conjugates and evaluation of their in-vivo toxicity are underway.  This is now reflected in the manuscript.

  • Page 6: Biodistribution of 64Cu-AuNPs 1 and 2 are missing.

We greatly appreciate reviewer’s comment. In our studies, we selected to examine biodistribution for those Au NP conjugates which showed high tumor uptake. Based on PET imaging results presented in the figure 2, Au NP conjugates 4-6 exhibit better tumor uptake compared to that of Au NP conjugates 1-3. Therefore, we did not carry out biodistribution studies for Au NP conjugates 1-3. This is now reflected in the text.

  • Page 9: Each ligand was added in a ratio of 900 to AuNPs. How it is guaranteed that both ligands are bound equally to the NPs?

We thank the reviewer for this valuable question. In order to precisely conjugate different types of ligands onto surface of Au NPs with pre-determined amounts, we have developed a novel method of sequential conjugation as reported in references 19, 24, and 28. By applying this method in the current study, we could control that both ligands are bound equally to the Au NPs and the amount of each ligand loaded is 900. This is now better reflected in the text.

  • Page 9 section 3.6: The QC of the product is missing as well as its radiochemical purity.

We greatly appreciate the reviewer for this comment. We have added radiochemical purity (line 397, page 10). Labeled 64Cu-AuNPs were washed with water twice to remove free 64Cu. We have tested the efficiency of the washing process and found that the activity recovered from 1st and 2nd wash were 88±0.4% and 97±0.02%, means greater than 97% of activity were attached on nanoparticles.  

  • Page 10 line 380: The PET scan times of 22 and 45 hours were based on what?

We thank the reviewer for this valuable question. Those time points were based on our scanner schedule which was close to 1 day and 2 days post injection. This is now better described in the manuscript.

  • Page 10 line 392: “The modified radiolabelling method has proven to be highly reliable and presents benefits for potential clinical translation in multimodality imaging or theragnosis of cancer.” Please specify which labeling method do you refer to.

We greatly appreciate the reviewer for this comment. We have specified this labeling method as “chelator-free Cu-64 radiolabelling” in the revised manuscript (lines 439-440, page 11).

  • Page 10 line 393:”Both in vitro and in vivo studies demonstrated that tumor uptakes of dual-ligand AuNPs was higher than that of single-ligand AuNPs”. I do not agree with the statement since no statistical analysis was shown. Please reformulate your conclusion more accurately.

We greatly appreciate reviewer’s comment. In the revised manuscript, we have added more PET images (figure 2 and 3) and SUV data (table 4). In addition, the conclusions have been revised.

  • In my opinion one crucial compound, 64Cu-AuNPS, is missing in the in vivo study since the HeLa 229 xenografts are known to be highly vascularized there is no way to prove that this increased uptake is not prefusion mediated.

We greatly appreciate reviewer for this comment. In cell experiments, the negative control with AuNPs containing only mPEG were tested, which showed only 0.10±0.02% uptake (line 215, page 6), so we did not include a negative control in animal study. In addition, the differences shown between single-ligand modified AuNPs and dual-ligand modified AuNPs provide evidence that increased uptake is not prefusion mediated or at least is not totally prefusion mediated.

Reviewer 3 Report

Several PET probes were synthesized using the method previously developed by the authors, and the uptake of the probes into tumor tissues were compared. Their compatible PET probes are conjugates of some functional ligands bound on gold-nano-particle core, and labeled by 64Cu. The biodistributions of the probes can varied with the combination of ligands assembled on. This reviewer thinks that the purpose describe in the introduction of the paper was almost achieved, however the most description in the conclusion were just their hopes rather the concrete fact based on evidences and data. There was no data of multimodality imaging in this paper. No therapeutic results were observed.

Author Response

Several PET probes were synthesized using the method previously developed by the authors, and the uptake of the probes into tumor tissues were compared. Their compatible PET probes are conjugates of some functional ligands bound on gold-nano-particle core, and labeled by 64Cu. The biodistributions of the probes can varied with the combination of ligands assembled on. This reviewer thinks that the purpose describe in the introduction of the paper was almost achieved, however the most description in the conclusion were just their hopes rather the concrete fact based on evidences and data. There was no data of multimodality imaging in this paper. No therapeutic results were observed.

We thank the reviewer for the statement about the merits of our work and the valuable comments for improving the manuscript. We agree with the reviewer and have added more PET images (figure 2 and 3) and SUV data (table 4). Also, the conclusions have been revised.

Round 2

Reviewer 2 Report

Review of manuscript pharmaceuticals-2000711-V2

The revised manuscript has answered most of my questions and after a few refinements, it will be suitable for publication.

Here are my final remarks (in green):

  • Page 5 line 182: “The fabricated AuNP conjugates were first evaluated in vitro using HeLa cells, the most commonly used human cancer cell line”. The most commonly used human cancer cell line in what? Why. Please explain.

We thank the reviewer for this question. We have explained in the revised manuscript that the HeLa cells are the most commonly used cancer cell line in scientific research because they are remarkably durable and prolific (lines 192-194, page 5). We have also provided a reference (44) for this explanation.

Perhaps the most important reason is that HeLa cells are cells known to express highly the folate receptors

  • Page 6 line245:” The replacement of mPEG with the glucose-modified PEG maintained the stability and biocompatibility of AuNPs…”How was this confirmed??

We thank the reviewer for this valuable question. Our statement that the replacement of mPEG with the glucose-modified PEG maintained the stability and biocompatibility of Au NPs was confirmed by not observing visible aggregation during manufacture and radiolabeling of glucose-modified AuNPs. This is now reflected in the manuscript.

My question was more related to their stability in vivo. But since you are only referring to the increased stability of the Glu-AuNP Vs PEG-AuNP in general I suggest removing the new sentence that was added on page 7 line 265 since it was already stated in the stability assay.

  • Page 6: The authors mention that the 64Cu-AuNPs could also be used in radiation therapy as well. With so high liver uptake is this feasible?

We thank the reviewer for pointing this out. Gold has been used as medicine for hundreds of years. Its metabolic mechanism is complicated and has not been fully understood yet. However, it has been known that coating AuNPs with PEG definitely reduces their toxicities and increases biocompatibility. Also, further modification of our Au NP conjugates and evaluation of their in-vivo toxicity are underway.  This is now reflected in the manuscript.

Although the AuNPs are not that toxic and they themselves may not cause toxicity, when attached to a radionuclide and, based on your in vivo study, I am certain that being exposed to such a high radiation dose in the liver most likely will cause toxicity to the patient. Therefore, I   suggest leaving the radiation therapy application out.

  • Page 6: Biodistribution of 64Cu-AuNPs 1 and 2 are missing.

We greatly appreciate reviewer’s comment. In our studies, we selected to examine biodistribution for those Au NP conjugates which showed high tumor uptake. Based on PET imaging results presented in the figure 2, Au NP conjugates 4-6 exhibit better tumor uptake compared to that of Au NP conjugates 1-3. Therefore, we did not carry out biodistribution studies for Au NP conjugates 1-3. This is now reflected in the text.

The way it is now written is very confusing thus I suggest changing it to ”…. As shown in Figures 2 and 3, the tumor uptake of dual ligand AUNPs 4,5, and 6 was higher than that of the single-ligand AUNPs 1,2 and 3. This is in marked contrast to PET imaging of free 64CuCl2, which shows rapid renal clearance (data not shown), and imaging of non-conjugated AuNPs which showed most activity in the liver within 5 minutes [25]. Furthermore, the glucose-modified AuNP conjugate 5 and 6 showed substantial improvement of tumor uptake….”

Author Response

Reviewer 2

The revised manuscript has answered most of my questions and after a few refinements, it will be suitable for publication.

Here are my final remarks (in green):

  • Page 5 line 182: “The fabricated AuNP conjugates were first evaluated in vitro using HeLa cells, the most commonly used human cancer cell line”. The most commonly used human cancer cell line in what? Why. Please explain.

We thank the reviewer for this question. We have explained in the revised manuscript that the HeLa cells are the most commonly used cancer cell line in scientific research because they are remarkably durable and prolific (lines 192-194, page 5). We have also provided a reference (44) for this explanation.

Perhaps the most important reason is that HeLa cells are cells known to express highly the folate receptors

Response: good point. We have also added this statement accordingly.

  • Page 6 line245:” The replacement of mPEG with the glucose-modified PEG maintained the stability and biocompatibility of AuNPs…”How was this confirmed??

We thank the reviewer for this valuable question. Our statement that the replacement of mPEG with the glucose-modified PEG maintained the stability and biocompatibility of Au NPs was confirmed by not observing visible aggregation during manufacture and radiolabeling of glucose-modified AuNPs. This is now reflected in the manuscript.

My question was more related to their stability in vivo. But since you are only referring to the increased stability of the Glu-AuNP Vs PEG-AuNP in general I suggest removing the new sentence that was added on page 7 line 265 since it was already stated in the stability assay.

Response: ok – we have removed it.

  • Page 6: The authors mention that the 64Cu-AuNPs could also be used in radiation therapy as well. With so high liver uptake is this feasible?

We thank the reviewer for pointing this out. Gold has been used as medicine for hundreds of years. Its metabolic mechanism is complicated and has not been fully understood yet. However, it has been known that coating AuNPs with PEG definitely reduces their toxicities and increases biocompatibility. Also, further modification of our Au NP conjugates and evaluation of their in-vivo toxicity are underway.  This is now reflected in the manuscript.

Although the AuNPs are not that toxic and they themselves may not cause toxicity, when attached to a radionuclide and, based on your in vivo study, I am certain that being exposed to such a high radiation dose in the liver most likely will cause toxicity to the patient. Therefore, I   suggest leaving the radiation therapy application out.

Response: We had edited the following sentence, and hope this is acceptable to the reviewer:

“Thus, this unique nano-platform offers a versatile tool for multimodality imaging and therapy methods: chemo-, thermal- and radiation therapies, although we recognize the latter application will only be possible if hepatic uptake can be substantially reduced so as to avoid liver damage from high doses of therapeutic radionuclides.”

  • Page 6: Biodistribution of 64Cu-AuNPs 1 and 2 are missing.

We greatly appreciate reviewer’s comment. In our studies, we selected to examine biodistribution for those Au NP conjugates which showed high tumor uptake. Based on PET imaging results presented in the figure 2, Au NP conjugates 4-6 exhibit better tumor uptake compared to that of Au NP conjugates 1-3. Therefore, we did not carry out biodistribution studies for Au NP conjugates 1-3. This is now reflected in the text.

The way it is now written is very confusing thus I suggest changing it to ”…. As shown in Figures 2 and 3, the tumor uptake of dual ligand AUNPs 4,5, and 6 was higher than that of the single-ligand AUNPs 1,2 and 3. This is in marked contrast to PET imaging of free 64CuCl2, which shows rapid renal clearance (data not shown), and imaging of non-conjugated AuNPs which showed most activity in the liver within 5 minutes [25]. Furthermore, the glucose-modified AuNP conjugate 5 and 6 showed substantial improvement of tumor uptake….”

Response: Changed as requested.

Reviewer 3 Report

This reviewer thinks that the authors just cosmetically revised their conclusion, however the most description in the conclusion were still their hopes rather the concrete fact based on any evidence and data. This reviewer could not see the "new pathway to combine multimodality imaging" on their PET probes, due to that there was still no data showing any possibility of "multimodality imaging" in this paper. This reviewer could not see no therapeutic or theragnosic results in this paper neither.

Author Response

Reviewer 3

This reviewer thinks that the authors just cosmetically revised their conclusion, however the most description in the conclusion were still their hopes rather the concrete fact based on any evidence and data. This reviewer could not see the "new pathway to combine multimodality imaging" on their PET probes, due to that there was still no data showing any possibility of "multimodality imaging" in this paper. This reviewer could not see no therapeutic or theragnosic results in this paper neither.

Response: Updated the conclusion and dialed back some of the claims as requested.